# Revisiting Learning Rate Control

Micha Henheik[1]   Theresa Eimer[1]   Marius Lindauer[1]

[1]Leibniz University Hannover

**Abstract**   The learning rate is one of the most important hyperparameters in deep learning, and how to control it is an active area within both AutoML and deep learning research. Approaches for learning rate control span from classic optimization to online scheduling based on gradient statistics. This paper compares paradigms to assess the current state of learning rate control. We find that methods from multi-fidelity hyperparameter optimization, fixed-hyperparameter schedules, and hyperparameter-free learning often perform very well on selected deep learning tasks but are not reliable across settings. This highlights the need for algorithm selection methods in learning rate control, which have been neglected so far by both the AutoML and deep learning communities. We also observe a trend of hyperparameter optimization approaches becoming less effective as models and tasks grow in complexity, even when combined with multi-fidelity approaches for more expensive model trainings. A focus on more relevant test tasks and new promising directions like finetunable methods and meta-learning will enable the AutoML community to significantly strengthen its impact on this crucial factor in deep learning.

## 1 Introduction

Deep learning produces state-of-the-art algorithms in applications such as image recognition (Z. Li and Ren, 2020; Dosovitskiy et al., 2021; Wortsman et al., 2022; Gu et al., 2024; Muszynski et al., 2024) and natural language processing (Vaswani et al., 2017; Brown et al., 2020). However, deep learning models are sensitive to their hyperparameters, including the learning rate of the optimizer. It is perhaps the hyperparameter most responsible for training success in deep learning (Goodfellow et al., 2016). Thus, effectively controlling the learning rate is a major factor for training success.

Automated Machine Learning (AutoML; Hutter, Kotthoff, et al., 2019) aims to automate the configuration of machine learning algorithms in an efficient manner, e.g. through black-box optimization combined with efficient scheduling strategies (multi-fidelity HPO; Jamieson and Talwalkar, 2016; Falkner et al., 2018; Bischl et al., 2023). Its adoption for hyperparameter optimization in the deep learning community, however, remains somewhat low (Bouthillier and Varoquaux, 2020). A key drawback of many approaches from the AutoML community is the need for repeated model training, which can become infeasibly expensive for complex deep learning tasks.

To alleviate this computational cost, hyperparameter-free optimization has emerged as an alternative to standard AutoML approaches, eliminating the need for external tuning by developing optimization methods without hyperparameters to adapt (Orabona and Cutkosky, 2020; Defazio and Mishchenko, 2023; Carmon and Hinder, 2024). Unlike hyperparameter schedules, which depend solely on the current optimization step (Loshchilov and Hutter, 2017; Smith and Topin, 2017), the learning process is used to determine the learning rate, e.g. via the development of the gradients. These approaches predominantly focus on the learning rate (Orabona and Tommasi, 2017; Defazio and Mishchenko, 2023; Carmon and Hinder, 2024), though there are also hyperparameter-free methods for other hyperparameters like the momentum (Levy et al., 2021). Naturally, this presents a significant cost reduction, especially in computationally intensive domains like language modeling.

We compare both multi-fidelity HPO and hyperparameter-free approaches to learning rate schedules on different deep learning tasks from *logistic regression*, *computer vision*, and *language*

*modeling*. We find that all options perform only situationally well, and none are a viable default choice across tasks. While multi-fidelity HPO performs well on the lightweight LIBSVM (Chang and Lin, 2011) benchmark, more complex deep learning problems favor varying hyperparameter-free methods. Therefore, our results identify a significant gap in learning rate control research: selecting between different configuration approaches and paradigms. Furthermore, they demonstrate that we cannot infer the superiority of multi-fidelity HPO approaches compared to hyperparameter-free methods from small deep-learning tasks. To increase the impact of AutoML in this important area of deep learning, we need to evaluate the use cases we want to target and embrace approaches like hyperparameter-free learning with strengths specific to expensive deep learning settings – here the lack of repeating runs. Our evaluation shows that there is still plenty of room for improvement over the overall best methods and thus fertile ground for novel AutoML research.

## 2 Prior Work on Learning Rate Control

We give a brief overview of important strands of learning rate control literature from the communities concerning our comparison: configuration approaches from the AutoML community, learning rate schedules, and hyperparameter-free configuration methods motivated by deep learning theory. The selection of optimizers themselves is related to our topic of learning rate control as well. Schmidt et al., 2020 conduct a similar investigation to ours for optimizers and also find algorithm selection to be a key challenge for that problem.

### 2.1 The AutoML Approach To Learning Rate Selection & Control

In the realm of AutoML, there are several paradigms to optimize hyperparameters for maximum performance, usually through repeated evaluations of the target function. To improve over simple baselines such as Random Search and Grid Search (Bergstra and Bengio, 2012), HPO methods commonly use either model-based or model-free optimization. Model-based configuration approaches (Hutter, Hoos, et al., 2011; Snoek et al., 2012) use a surrogate model of the target algorithm's performance to guide the search for good configurations with *Bayesian Optimization* (Brochu et al., 2010). Model-free methods, on the other hand, rely on paradigms such as evolutionary strategies to evolve well-performing configurations over time (Ansótegui et al., 2009; X. He et al., 2021).

To improve the efficiency of these methods, partial target function evaluations are employed to eliminate bad configurations early on (L. Li et al., 2018). Such scheduling approaches are referred to as *multi-fidelity* methods and common in both model-based and model-free HPO (Falkner et al., 2018; Awad et al., 2021). Since these methods can make HPO approaches based on black-box optimization significantly more efficient, we focus on multi-fidelity HPO in this comparison.

The majority of HPO approaches target finding a single configuration for the full training run. Dynamic algorithm configuration (DAC; Adriaensen et al., 2022) is a recent paradigm adapting hyperparameters during the run and it has been applied to simple deep learning tasks (Daniel et al., 2016), but so far, there are no standard DAC solvers. Therefore, we do not include DAC in our comparison and instead focus on approaches deep learning practitioners can apply out of the box.

Configuring deep learning algorithms is part of several AutoML benchmarks like HPOBench (Eggensperger et al., 2021), LCBench (Zimmer et al., 2021), or PD1 (Wang et al., 2024). Except for PD1, however, these benchmarks are quite limited compared to deep learning in state-of-the-art systems, using far smaller networks and less complex datasets than current research. PD1 is a new addition with relevant architectures like ResNets (K. He et al., 2016) and Transformers (Vaswani et al., 2017), though currently most AutoML literature around learning rate control is still centered around applying methods for HPO to relatively simple deep learning tasks.

### 2.2 Learning Rate Schedules

In contrast to multi-fidelity HPO approaches, learning rate *schedules* are not based on optimization but pre-defined heuristic approaches that do not require multiple runs. Usually, they are a function

of the current time step, adapting the learning rate during training time. Popular choices are Step Decay (Ge et al., 2019), Exponential Decay (Z. Li and Arora, 2020) and *Cosine Annealing Warm Restarts* (CAWR, Loshchilov and Hutter, 2017). CAWR sets the learning rate in step $t$ according to

$$\eta_t = \eta_{\min}^i + \frac{1}{2}\left(\eta_{\max}^i - \eta_{\min}^i\right)\left(1 + \cos\left(\frac{T_{\mathrm{cur}}}{T_i}\pi\right)\right)$$

where $i$ is the restart counter, $\eta_{\min}$ and $\eta_{\max}$ are bounds on the learning rate and $T_{\mathrm{cur}}$ the number of steps since the last restart. Intuitively, this means that the learning rate decays from $\eta_{\max}$ to $\eta_{\min}$ in a cosine shape in $T_i$ steps. After that, it resets to $\eta_{max}$ and scales $T_i$ with $T_{\mathrm{mult}}$. While computationally inexpensive compared to HPO, such schedules often make use of several hyperparameters. Fixed schedules cannot adapt to different settings and thus possibly do not work well when moving to a different target algorithm or dataset.

## 2.3 From Theory: Hyperparameter-free Learning

Researchers have looked to deep learning theory to combine the efficiency of learning rate schedules with the adaptiveness of multi-fidelity HPO approaches. Setting the optimal learning rate of deep learning optimizers is possible using information about the problem, e.g. distance $D$ between the initial iterate and the optimum (Defazio and Mishchenko, 2023). However, this information is usually not directly available. D-Adaptation (Defazio and Mishchenko, 2023) and Prodigy (Mishchenko and Defazio, 2024) estimate a lower bound on the distance $D$ to the optimum via gradient statistics. Prodigy is an update on the original D-Adaptation with improved convergence rate and more exploitative behavior. Similarly, *Distance over Weighted Gradients* (DoWG; Khaled et al., 2023) computes the step size of SGD using this simple estimate of $D$ and a weighted sum of observed Gradients. COCOB (Orabona and Tommasi, 2017), on the other hand, reduces the learning rate control task to a coin betting scenario. COCOB uses a betting strategy to solve coin betting and achieves optimal convergence for this scenario. However, this optimality is specific to assumptions and may not extend to deep learning tasks.

While these theory-motivated approaches promise to be the zero-shot solution for learning rate control and have shown impressive performances in their own evaluations, there is as of yet no broad comparison of hyperparameter-free learning approaches, let alone how they fare compared to HPO methods. In fact, there has been discussion about the fairness of evaluations for some hyperparameter-free learning methods (Orabona, 2023). Thus, we want to validate how the different strands of hyperparameter-free learning compare to each other and to learning rate schedules and multi-fidelity HPO methods, establishing common ground between these communities.

## 3 Comparing Learning Rate Control Approaches

In this section, we describe our empirical evaluation of three different learning rate control approaches: we evaluate *hyperparameter-free* methods alongside *multi-fidelity HPO* and *scheduling approaches* on a diverse set of tasks. Our experiments span domains including natural language processing, computer vision, and convex optimization. The full details on the datasets and corresponding architectures can be found in Appendix A.

**Hyperparameter-free Methods** Our study includes four hyperparameter-free methods: *COCOB*, a method that reduces learning rate control to a simpler problem, and *D-Adaptation*, *Prodigy*, and *DoWG*, which are methods that estimate the optimal learning rates through gradient information. As these methods are proposed as tuning-free, i.e. they do not possess any important hyperparameters to tune, we apply them directly in all experiments.

**Multi-fidelity HPO** For multi-fidelity HPO, we use SMAC3 (Lindauer et al., 2022), a state-of-the-art HPO tool (Eggensperger et al., 2021), with the Hyperband (L. Li et al., 2018) intensifier to find the best fixed learning rate, i.e. the learning rate multi-fidelity HPO finds stays constant during

the whole run. We call this approach *SMAC*. SMAC's total optimization budget is 50 trials for all experiments. Hyperband is used with a minimum budget depending on the task (see Appendix A), a maximum budget equal to the total number of iterations, and $\eta = 3$.

**Scheduling Approaches** We focus on *Cosine Annealing* (Loshchilov and Hutter, 2017) as a learning rate schedule due to its impressive performance and adjustment options. These adjustment options allow us to report both the performance of cosine annealing with its default settings and a SMAC-tuned variation that shows how much we can adapt the schedule to new settings. We use the cosine annealing as originally proposed with warm restarts for the entire empirical evaluation. As *Cosine Annealing Warm Restarts (CAWR)* default, we use a well-performing configuration ($T_0 = 10, T_{mult} = 2, \eta_{min} = 0, \eta_{max} = 0.005$) from Loshchilov and Hutter, 2017 that achieved great results on CIFAR-10 and CIFAR-100. *Tuned CAWR* denotes runs where we use SMAC3 to optimize CAWR's four hyperparameters $T_0, T_{mult}, \eta_{min}, \eta_{max}$ (see Appendix A).

### 3.1 Experimental Setup

As an experimental framework, we employ the DACBench (Eimer et al., 2021) *SGD Benchmark*, which provides a structured environment for evaluating hyperparameter scheduling approaches. We evaluate all methods on deep learning settings from different domains: i. As a simple yet diverse application, we consider *logistic regression* in the form of the LIBSVM (Chang and Lin, 2011) library on nine of its well-studied datasets. ii. medium complexity *computer vision* taks with ResNet (K. He et al., 2016) variations on the CIFAR-10 and CIFAR-100 (Krizhevsky, 2009) datasets and a ViT (Dosovitskiy et al., 2021) on the Describable Textures Dataset (DTD; Cimpoi et al., 2014). iii. in *natural language processing*, we pre-train *RoBERTa* (Liu et al., 2019) architecture on a reproduction of the BookWiki dataset[1]. As DoWG does not support weight decay, we do not use it. Unless otherwise specified, we use the beta defaults $\beta_1 = 0.9, \beta_2 = 0.999$. For other hyperparameters and number of seeds, we use problem-specific standards with details in Appendix A. Our full code is available at `https://github.com/automl/Revisiting_LR_Control`. Our raw experiment data with the results of all runs is available on HuggingFace at `https://huggingface.co/datasets/autorl-org/revisiting_learning_rate_control`.

**Logistic Regression** We evaluate on the nine datasets Aloi, Dna, Iris, Letter, Pendigits, Sensorless, Vehicle, Vowel and Wine (Anderson and Goldenstein, 2014; Dua and Graff, 2017). They have been included in several works on hyperparameter-free learning and are domains where these methods perform well compared to their own baselines (Defazio and Mishchenko, 2023; Orabona, 2023; Mishchenko and Defazio, 2024). As there do not exist typical splits and some datasets are very small, we only report training loss and accuracy. Every method is executed for 100 epochs.

**Computer Vision** We train WRN-16-8 (Zagoruyko and Komodakis, 2016) on CIFAR-10 and DenseNet-121 (Huang et al., 2017) for CIFAR-100 to test two variations of ResNet architectures. On DTD we use the ViT/T-16 (Dosovitskiy et al., 2021). With CIFAR and DTD, we evaluate learning rate control approaches on standard task settings and architectures for many practical deep learning applications. We run these experiments for 300 epochs.

**Natural Language Processing** On BookWiki with RoBERTa we use $\beta_2 = 0.98$ and a learning rate warmup of 10 000 steps with default linear decay. For *COCOB* and *DoWG*, adopting the decay involves scaling the gradients by the learning rate. For *CAWR* and *Tuned CAWR*, the cosine schedule takes over after warmup. We limit the experiment to 23 000 steps.

### 3.2 LIBSVM Classification Datasets

We begin with a comparison of learning rate control approaches on a selection of simple but diverse classification tasks from LIBSVM datasets. Our results reflect this diversity (see Figure 2

---

[1]We try to replicate BookWiki from Defazio and Mishchenko, 2023 which is not publicly available. Therefore, we use a snapshot from Wikipedia concatenated with books from Zhu et al., 2015.

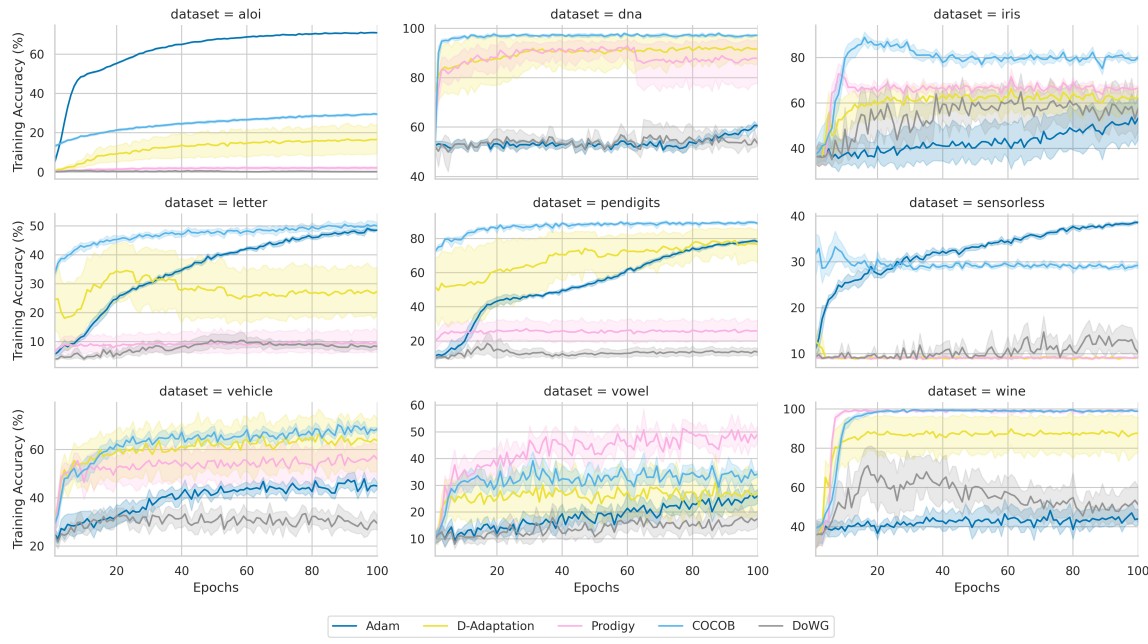

Figure 2: Training Accuracies for *hyperparameter-free* methods on LIBSVM datasets. Figures show the mean across 10 seeds with standard error.

for hyperparameter-free variations): COCOB, Prodigy and the default learning rate perform best on at least one dataset, and each method performs below a training accuracy of 50% at least once.

In terms of overall performance, COCOB performs best among the hyperparameter-free methods, setting or matching the best training accuracy in six out of nine cases. Prodigy and the default learning rate are inconsistent, sometimes not improving training accuracy at all (e.g. on *aloi* for Prodigy or *wine* for the default). D-Adaptation varies less across the datasets but only matches the top performance once on *vehicle*. DoWG does not perform well on any dataset, being by far the worst choice of the methods we tested. Therefore, we can conclude that none of the hyperparameter-free approaches work well overall.

These inconclusive results are less pronounced for HPO approaches (see Figure 3). CAWR in its untuned variation outperforms the default static learning rate, but cannot find a good solution on *wine*. Its tuned counterpart outperforms default CAWR on some datasets, solving *wine*, but is also less stable and thus suboptimal on others like *sensor*. SMAC performs best among HPO approaches, improving upon the previous best, COCOB. In fact, SMAC is the

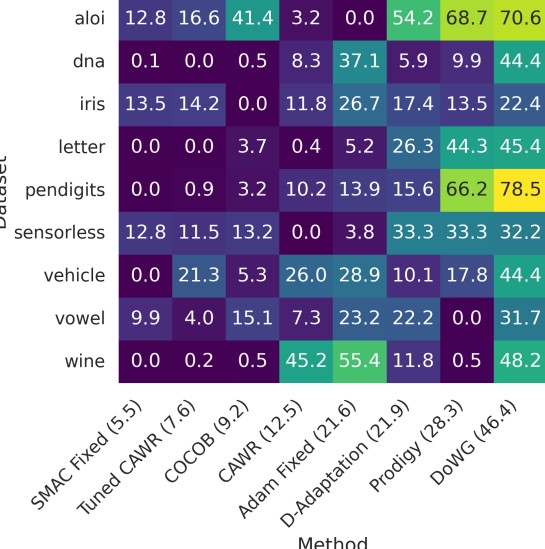

Figure 1: Average differences in final training accuracy of every method to the oracle on each LIBSVM dataset. Methods are sorted according to the mean average difference across all datasets (in parentheses).

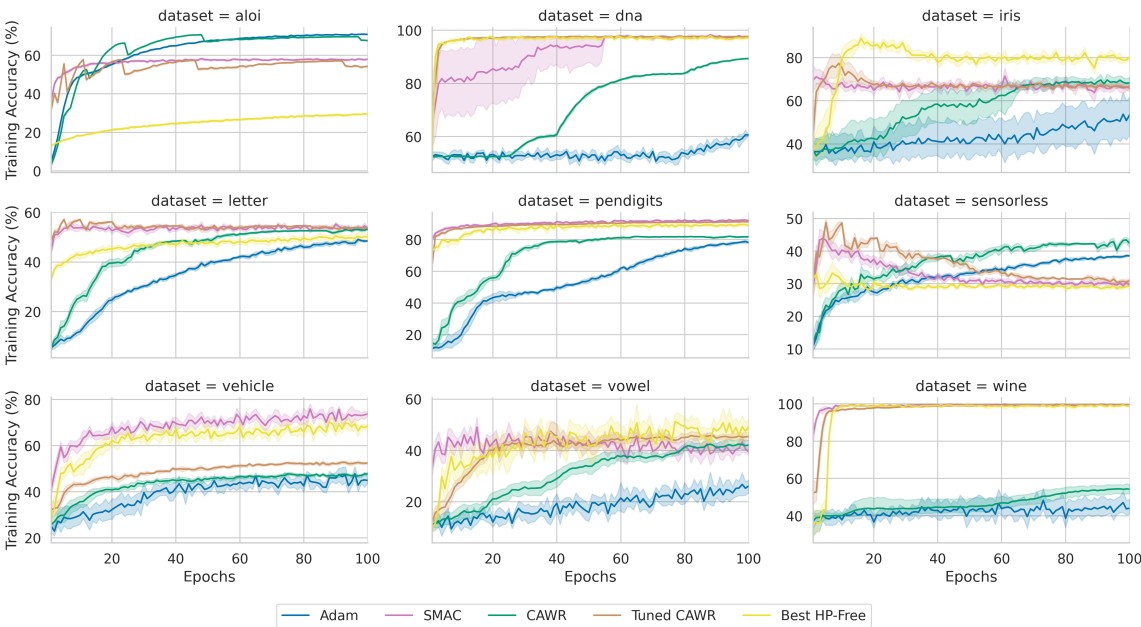

Figure 3: Training Accuracies for LIBSVM, *non-hyperparameter-free* methods. Figures show the mean across 10 seeds with standard error. We add the best hyperparameter-free method according to the final training accuracy. The Best HP-Free method is COCOB for every dataset except vowel and wine. For the latter it is Prodigy.

best overall method in this comparison, ranking first or second on every dataset. Thus, it is the only consistent approach, with an overall mean difference to the optimum of only 5.5% (see Figure 1).

For optimal performance across these simple datasets, our results show that we need a portfolio of six different methods (their marginal accuracy contribution in brackets): SMAC (6.8%), Tuned CAWR (0.1%), COCOB (11.8%), CAWR (3.8%), the default learning rate (3.2%) and Prodigy (4%)). Even ignoring Tuned CAWR due to its small contribution leaves us with a portfolio of a multi-fidelity HPO tool, a schedule, two hyperparameter-free learning methods, and the default learning rate.

## 3.3 Computer Vision Tasks

The LIBSVM tasks are not very representative of the currently most interesting tasks for learning rate control. Therefore, we repeat this evaluation on CIFAR-10, CIFAR-100, and DTD as archetypal deep learning tasks of medium complexity. Figures 4 and 5 show the results for DTD and CIFAR-100, respectively. Since the learning curves of CIFAR-10 and CIFAR-100 show similar trends, we only show CIFAR-100 here; refer to Appendix C for the full results.

On these datasets, hyperparameter-free methods perform well in terms of validation accuracy, even though the validation loss for DOWG on CIFAR-100 diverges. DoWG produces adequate, if not competitive, results compared to its poor performance on the LIBSVM datasets. COCOB's relative performance is worse, not being able to compete with other methods on the CIFAR datasets, even though it does well on DTD. Prodigy excels on these tasks, clearly outperforming D-Adaptation, and thus showing that its exploitative behavior is helpful in these settings.

Performance of HPO methods (see Figure 5) also changes from LIBSVM. We can now clearly see that the default CAWR schedule, while quite successful on the LIBSVM data, causes significant performance drops upon reset, leading to less stable performance and, on CIFAR-100 and DTD, to a clear performance drop compared to other methods. The tuned variation is much smoother and performs on par with the default learning rate on CIFAR, though it fails to learn on DTD.. SMAC cannot match it on CIFAR-100 with a gap of 2.79%, but matches the default learning rate's good

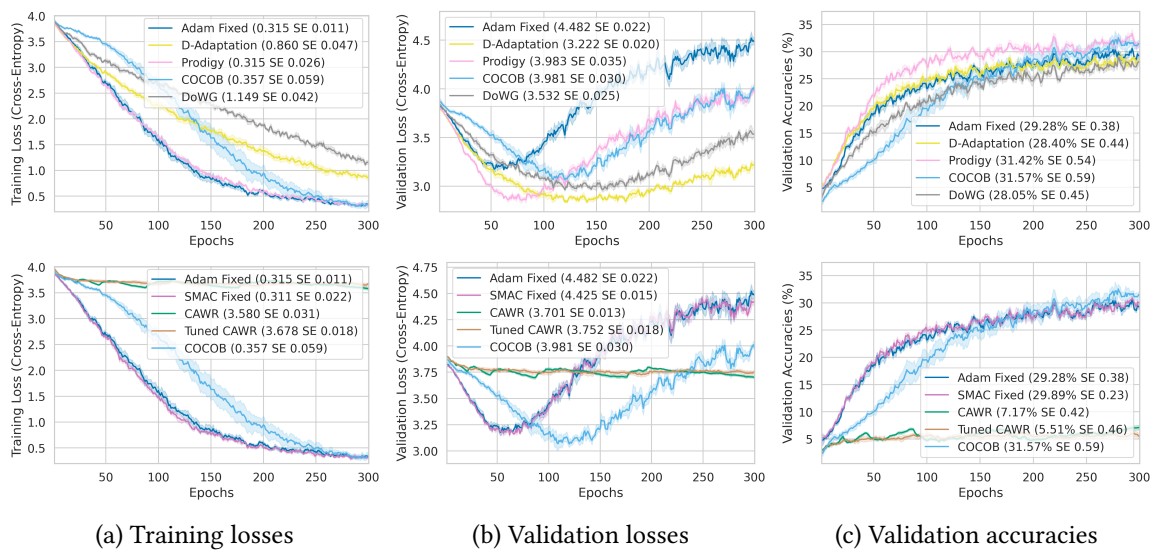

(a) Training losses        (b) Validation losses       (c) Validation accuracies

Figure 4: Resulting Training losses (a), Validation losses (b) and Validation Accuracies (c) for the *hyperparameter-free* methods (first row) and *non-hyperparameter-free* methods (second row) on DTD. In the second row, the best hyperparameter-free method (COCOB) is added to the plot. Figures show the mean across seeeds with standard error.

| | Adam | SMAC | CAWR | T. CAWR | D-Adapt | Prodigy | COCOB | DoWG |
|---|---|---|---|---|---|---|---|---|
| **CIFAR-10** | 0.06 | 0.61 | 0.71 | 0.24 | 0.41 | **0.0** | 1.20 | 0.54 |
| **CIFAR-100** | 0.80 | 3.59 | 7.93 | 1.40 | 1.86 | **0.0** | 4.72 | 6.00 |
| **DTD** | 2.28 | 1.68 | 24.40 | 26.05 | 3.16 | 0.15 | **0.0** | 3.51 |
| **Overall** | 1.05 | 1.96 | 11.01 | 9.23 | 1.81 | **0.05** | 1.97 | 3.35 |

Table 1: Average differences in final validation accuracy of every method compared to the oracle on computer vision datasets. All numbers are in %. Best method on each dataset is marked bold.

performance on DTD. It is not fully clear why SMAC cannot recover the default learning rate on CIFAR. We believe it is not due to multi-fidelity scheduling since the incumbent was selected at 100 epochs, after which no significant changes in performance happen for any method. It is more likely that the HPO landscape is harder to navigate, as preliminary results suggest that in deep learning, well-performing hyperparameters are close to regions of instability (Sohl-Dickstein, 2024).

We can see that the average gap to the optimum (see Table 1) is now smallest for Prodigy, with the default learning rate being ranked second, doing better than SMAC and other hyperparameter-free methods. For optimal performance, we only need a portfolio of COCOB and Prodigy for these datasets, with COCOB's marginal contribution being very low at 0.15%. CAWR cannot improve upon the default learning rate here and methods still show big discrepancies between datasets, though hyperparameter-free methods perform better overall here than on LIBSVM.

### 3.4 Language Model Training

We move to an even more complex deep learning task and an even larger model: pre-training RoBERTa with a one-cycle warmup and decay schedule for all learning rates, see Figure 6. We see that most methods deteriorate at some point during training to a local minimum. Only DoWG and D-Adaptation keep improving over time. This is surprising since both methods have not been among our best choices for any dataset so far. Figure 6c shows the likely cause for this performance collapse: D-Adaptation has by far the lowest learning rate and the methods to collapse earliest are

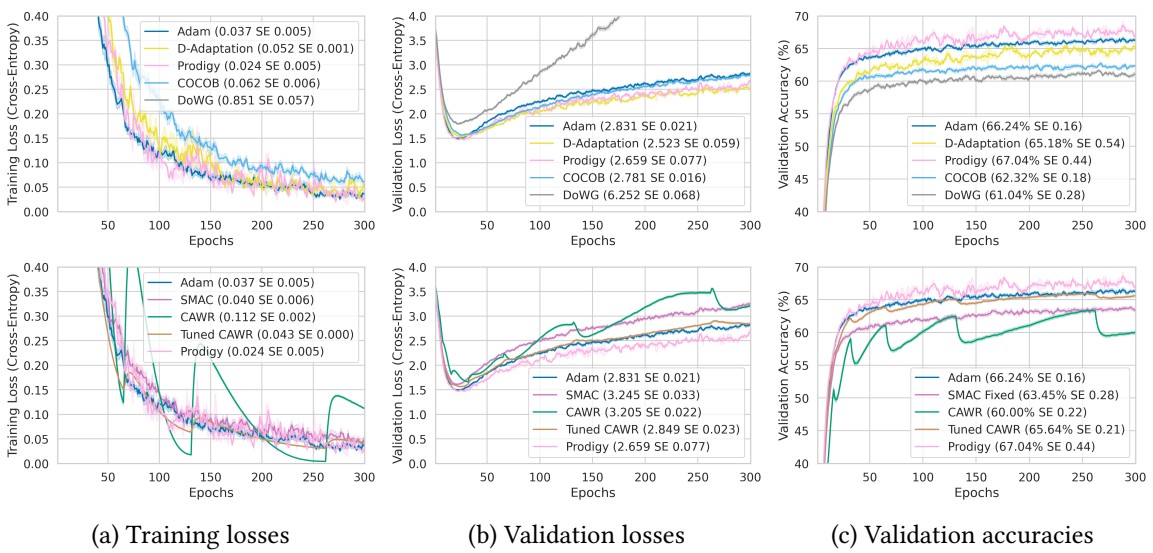

(a) Training losses  (b) Validation losses  (c) Validation accuracies

Figure 5: Resulting Training losses (a), Validation losses (b) and Validation Accuracies (c) for the *hyperparameter-free* methods (first row) and *non-hyperparameter-free* methods (second row) on CIFAR-100. In the second row, the best hyperparameter-free method is added to the plot. Figures show the mean across seeeds with standard error.

the methods with the highest learning rates, CAWR and SMAC. Since RoBERTa seems sensitive to larger learning rates, the resets in CAWR become a risky strategy. However, we can see that CAWR and SMAC are successful during the first optimization steps. We theorize this is why SMAC selects such a high learning rate. What SMAC sees as promising configurations on lower fidelities are unstable choices for full runs. This suggests that multi-fidelity optimization can be difficult to set up correctly for RoBERTa, even though these function evaluations are exceedingly costly.

The default learning rate and Prodigy collapse later, at around half of our optimization steps. Prodigy was conceived as a more exploitative version of D-Adaptation (Defazio and Mishchenko, 2023), and this change likely contributes to a worse performance in our evaluation. It also makes explicit that while Prodigy outperforms D-Adaptation on other domains, it is not a universal improvement but instead a different inductive bias. COCOB performs very poorly in this experiment with no notable improvement. Thus, our experiments have shown COCOB to progressively get worse with larger networks and more complex tasks. DoWG has shown the opposite trend, though even here its final validation perplexity is worse than D-Adaptation's by a gap of 18.14. Clearly, these methods also have a strong inductive bias and are not universal learning rate control mechanisms.

## 4 Conclusion: What Is Best For Learning Rate Control?

Our results point to a chaotic state within learning rate control. No method we tested generalizes particularly well across deep learning tasks. For an optimal solution of all our tasks, we need a portfolio of six methods. Even considering only the best method per domain, SMAC, Prodigy and D-Adaptation, we are left with a portfolio of three approaches to learning rate control with very different performance profiles, as shown by their marginal contributions of 0.69%, 0.07% and 0.11% respectively (see Table 2). We conclude that there are several excellent strategies for learning rate control, but that the problem is too varied to be solved by any one existing approach, even in the limited selection in our evaluation. Given the significant overhead of SMAC and other multi-fidelity HPO methods compared to hyperparameter-free learning, however, our results show that black-box HPO as a default choice is impractical for many deep learning settings. With the

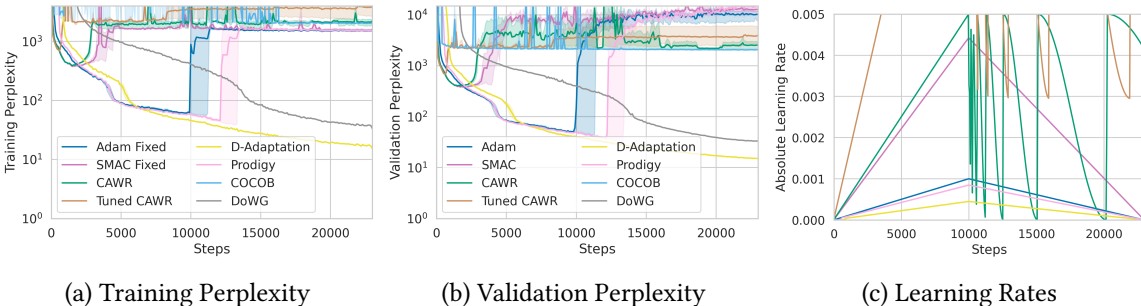

|  | (a) Training Perplexity | (b) Validation Perplexity | (c) Learning Rates |

Figure 6: Resulting Training Perplexity (a), Validation Perplexity (b) and observed Learning Rates (c) for the all methods on BookWiki using RoBERTa. The y-axes of (a) and (b) are on a log scale and show the mean across seeeds with standard error.

|  | Adam | SMAC | CAWR | T. CAWR | D-Adapt | Prodigy | COCOB | DoWG |
|---|---|---|---|---|---|---|---|---|
| **LIBSVM** ↑ | −0.39 | **1.97** | 0.96 | 1.65 | −0.40 | −1.31 | 1.41 | −3.88 |
| **CIFAR-10** ↑ | 0.06 | −0.02 | −0.03 | 0.03 | 0.01 | **0.07** | −0.10 | −0.01 |
| **CIFAR-100** ↑ | 0.34 | −0.06 | −0.55 | 0.25 | 0.19 | **0.45** | −0.22 | −0.40 |
| **DTD** ↑ | 0.77 | 0.85 | −2.39 | −2.63 | 0.64 | 1.07 | **1.09** | 0.59 |
| **Overall** ↑ | 0.20 | 0.69 | −0.50 | −0.18 | 0.11 | 0.07 | 0.55 | −0.93 |
| **BookWiki** ↓ | 627.8 | 985.2 | −464.2 | −273.7 | **−824.4** | 1303.2 | −532.1 | −821.8 |

Table 2: This table shows the marginal contribution to average portfolio accuracy or perplexity (on BookWiki) of all methods for every dataset. All values except for BookWiki are in percent (%). For accuracy higher is better and for perplexity lower is better. Best methods are marked bold.

unoptimized standard setting we test, it falls behind on complex tasks, and we see that its most efficient cost-saving measure, multi-fidelity optimization, can lead to poor performances.

That does not mean the AutoML community should stop focusing on learning rate control. More than anything, we show that selection between control mechanisms is extremely important. Furthermore, the tuned and default versions of CAWR show complementary strengths, showing that learning rate schedules benefit from tuning - and that, quite possibly, the same is true for hyperparameter-free methods. Right now, these methods claim to function fully tuning-free, even though we can clearly see that they cannot adapt to every task. Therefore, exposing a few key hyperparameters and focusing on tuning these for new tasks could be a best-of-both-worlds approach combining the concept behind hyperparameter-free learning and current HPO methods. To test these approaches, we believe evaluations on complex deep learning tasks, or the creation of surrogate versions like in NASBench-301 (Zela et al., 2022), are instrumental since the decreasing effect of multi-fidelity HPO methods is not obvious on the smaller datasets in our comparison.

We furthermore show hyperparameter-free learning methods to be brittle and situational, but to perform exceptionally well when suited to the task. Therefore, they can be used to meta-learn improved learning rate control strategies. Existing efforts have used reinforcement learning to learn dynamic control strategies for CMA-ES step sizes from existing heuristics (Shala et al., 2020). The same could be done for learning rate control, resulting in learned mechanisms that generalize better across settings. Without this problem of inconsistent performance, learning rate control mechanisms will be much more appealing to the broader ML community than current methods.

**Broader Impact** After careful reflection, the authors have determined that this work presents no notable negative impacts to society or the environment.

**Acknowledgements**. Theresa Eimer acknowledges funding by the German Research Foundation (DFG) under LI 2801/7-1. This project was supported by the Federal Ministry of Education and Research (BMBF) under the project AI service center KISSKI (grant no.01IS22093C).

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

# A Experimental Details

Table 3: LIBSVM Configuration

| Parameter | Value |
| --- | --- |
| Architecture | Logistic Regression |
| Epochs | 100 |
| CPUs | 1×AMD Epyc |
| Device Batch Size | 64 |
| LR schedule | None |
| Seeds | 10 |
| decay | 0 |
| $\beta_1, \beta_2$ | 0.9, 0.999 |

Table 4: CIFAR-10 Configuration

| Parameter | Value |
| --- | --- |
| Architecture | Wide ResNet 16-8 |
| Epochs | 300 |
| GPUs | 1×H100 |
| Device Batch Size | 64 |
| LR schedule | None |
| Seeds | 10 |
| decay | 0 |
| $\beta_1, \beta_2$ | 0.9, 0.999 |

Table 5: CIFAR-100 Configuration

| Parameter | Value |
| --- | --- |
| Architecture | DenseNet-121 |
| Epochs | 300 |
| GPUs | 1×H100 |
| Device Batch Size | 64 |
| LR schedule | None |
| Seeds | 10 |
| decay | 0 |
| $\beta_1, \beta_2$ | 0.9, 0.999 |

Table 6: RoBERTa BookWiki Configuration

| Parameter | Value |
| --- | --- |
| Architecture | roberta_base |
| Task | masked_lm |
| Max updates | 23,000 |
| GPUs | 4×H100 |
| Max Length | 512 |
| Dropout | 0.1 |
| Attention Dropout | 0.1 |
| Device Batch Size | 64 |
| Warmup | 10,000 |
| Fp16 | True |
| Gradient Accumulation | 1 |
| LR schedule | None |
| Seeds | 3 |
| Decay | 0.0 |
| Adam LR | 0.001 |
| $\beta_1, \beta_2$ | 0.9, 0.98 |

Table 7: DTD Configuration

| Parameter | Value |
|---|---|
| Architecture | ViT Tiny Patch 16_224 |
| Epochs | 300 |
| GPUs | 1×H100 |
| Device Batch Size | 64 |
| LR schedule | None |
| Seeds | 10 |
| decay | 0 |
| $\beta_1, \beta_2$ | 0.9, 0.999 |

All of our experiments are executed on a slurm cluster. For experiments utilizing a GPU. A full training run of CIFAR-10 or CIFAR-100 experiments uses one H100 GPU and takes about 5 hours. One run on DTD takes about 3 hours. With regard to number of methods and seeds as well as tuning runs (see Tabel 4,5,7), our computer vision experiments utilized approximately 770 GPU hours. One run of our RoBERTa experiments used 4×H100 and takes 6 hours. Therefore, the natural language processing experiments including tuning runs used approximately 610 GPU hours. In total, we invested approximately 1380 GPU hours for all experiments.

|  |  | LIBSVM | CIFAR-10 | CIFAR-100 | DTD | RoBERTa |
|---|---|---|---|---|---|---|
| **SMAC** | n_trials | | | 50 | | |
| | min_budget | 20 | 5 | 5 | 5 | 500 |
| | max_budget | 100 | 300 | 300 | 300 | 23 000 |
| | $\eta$ | | | 3 | | |
| | log | | | $false$ | | |
| | Searchspace | | | | | |
| | lr | [0, 1] | [0.1, 1] | [0.1, 1] | [0.1, 1] | [0, 1] |
| **Tuned CAWR** | n_trials | | | 50 | | |
| | min_budget | 20 | 3125 | 3125 | 145 | 500 |
| | max_budget | max batch steps | 187 500 | 187 500 | 8 700 | 23 000 |
| | $\eta$ | | | 3 | | |
| | log | | | $false$ | | |
| | Searchspace | | | | | |
| | $\eta_{min}$ | | | [0, 0.005] | | |
| | base_lr | [0, 1] | [0, 0.1] | [0, 0.1] | [0, 0.1] | [0, 1] |
| | $T_0$ | | | [0, 50] | | |
| | $T_{mult}$ | | | [1, 5] | | |

Table 8: This table shows the configuration of SMAC and Hyperband used to tune our multi-fidelity HPO methods. Notice that the search spaces for the learning rates on LIBSVM and for RoBERTa are set to a broader spectrum than the computer vision experiments as there are no immediate typical ranges.

Table 9: This table shows the learning rate and CAWR hyperparameters found by SMAC on LIBSVM. Additionally, we report the mean final adapted learning rate of D-Adaptation and Prodigy.

| Method | Aloi | Dna | Iris | Letter | Pendigits | Sensorless | Vehicle | Vowel | Wine |
|---|---|---|---|---|---|---|---|---|---|
| **SMAC** | 0.2708 | 0.9523 | 0.9863 | 0.7047 | 0.7532 | 0.7031 | 0.9394 | 0.9896 | 0.9896 |
| **D-Adaptation** | 1.7224 | 3.8347 | 5.4031 | 3.0777 | 4.5138 | 2.1328 | 2.5959 | 4.8939 | 0.0000 |
| **Prodigy** | 3.3224 | 4.1537 | 7.7059 | 3.0777 | 11.0018 | 10.4012 | 7.5469 | 6.9458 | 9.9160 |
| **Tuned CAWR** | | | | | | | | | |
| eta_min | 0.0030 | 0.0030 | 0.0043 | 0.0030 | 0.0030 | 0.0030 | 0.0011 | 0.0011 | 0.0043 |
| base_lr | 0.5987 | 0.5987 | 0.2405 | 0.5987 | 0.5987 | 0.5987 | 0.0180 | 0.0180 | 0.2405 |
| T_0 | 38 | 38 | 30 | 38 | 38 | 38 | 45 | 45 | 30 |
| T_mult | 2 | 2 | 4 | 2 | 2 | 2 | 5 | 5 | 4 |

| | DTD | CIFAR-10 | CIFAR-100 | BookWiki |
|---|---|---|---|---|
| **SMAC lr** | 0.000880923938 | 0.0193164317787 | 0.0022199946895 | 0.0044237631561 |
| **D-Adaptation lr** | 0.000029 | 0.001541 | 0.000245 | $3.453551 \times 10^{-8}$ |
| **Prodigy lr** | 0.000407 | 0.002246 | 0.001589 | $2.809611 \times 10^{-4}$ |
| *Tuned CAWR* | | | | |
| **eta_min** | 0.0012426552362 | 0.0048900232824 | 0.0049729217242 | 0.0029495649644 |
| **base_lr** | 0.0048815239592 | 0.0057869866522 | 0.0019257718469 | 0.0142293709729 |
| **T0** | 40 | 29 | 39 | 47 |
| **T_mult** | 1 | 4 | 2 | 2 |

Table 10: This table shows the learning rate and CAWR hyperparameters found by SMAC on computer vision and natural language processing experiments. Additionally, we report the mean final adapted learning rate of D-Adaptation and Prodigy.

## B  Additional Plots

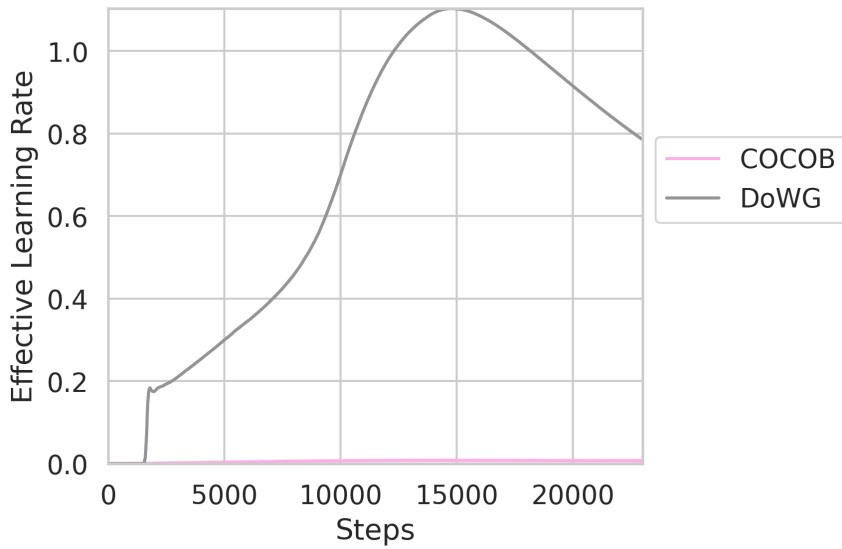

Figure 7: Estimated effective learning rates of DoWG and COCOB on the RoBERTa BookWiki experiments.

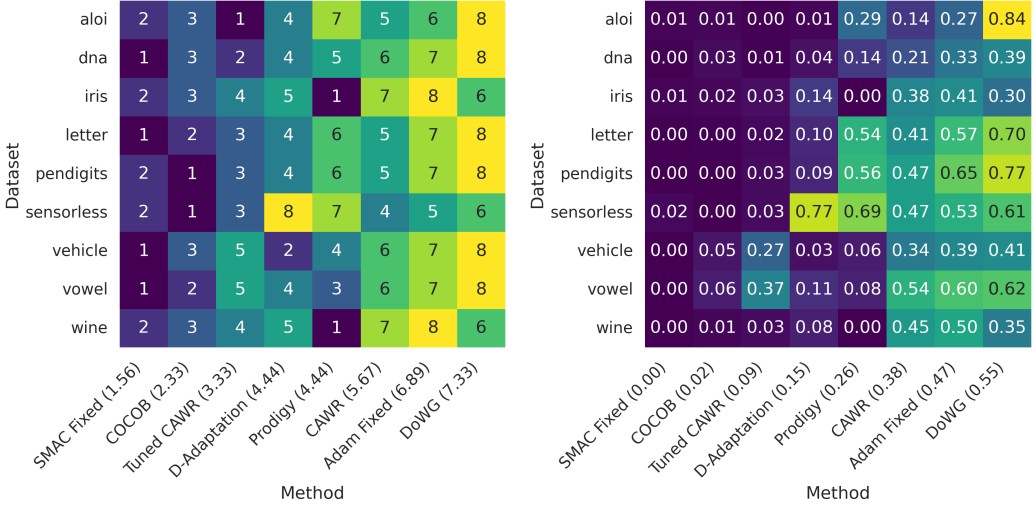

(a) Rank heatmap for the LIBSVM experiments according to final training loss. The average rank in denoted in parantheses.

(b) Average Difference Heatmap (Training Loss)

# C CIFAR-10 Results

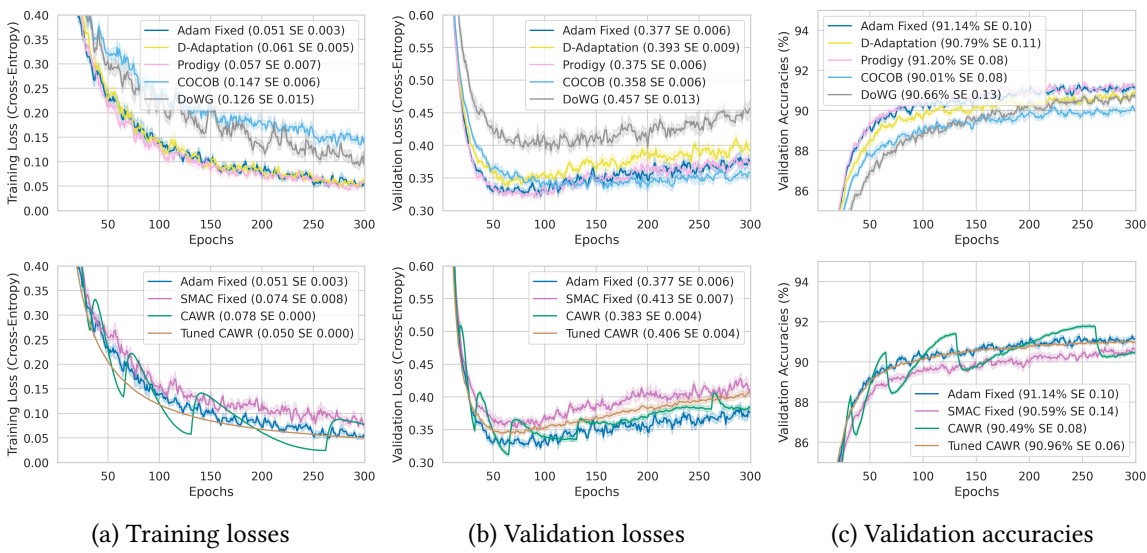

(a) Training losses      (b) Validation losses      (c) Validation accuracies

Figure 9: Resulting Training losses (a), Validation losses (b) and Validation Accuracies (c) for the *hyperparameter-free* methods (first row) and *non-hyperparameter-free* methods (second row) on CIFAR-10. In the first row, the best hyperparameter-free method is added to the plot.

