# OpenReview forum: "Revisiting Learning Rate Control"
_automl.cc/AutoML/2025/Methods_Track — AutoML 2025 Methods Track_

### Official Review · Reviewer_YnT7 · 2025-04-29

**Comments To Authors:**

This paper presents an extensive study of the performance of different learning rate adaptation techniques on different machine learning tasks. It includes hyperparameter-free methods (such as D-Adaptation), a multi-fidelity HPO method (SMAC), as well as a pre-defined learning rate schedule (cosine annealing). The study includes a variety of models (such as ResNets and Transformers) trained on tabular, computer vision, and natural language processing datasets. The main takeaway of the paper is that no method works best across all settings.

In general, the paper is well written, and the focus on the learning rate as arguably the most important hyperparameter in deep learning is reasonably motivated. While there have been papers in the past that compare different learning rates and learning rate schedules, the aspects explored in this study (hyperparameter-free vs. HPO vs. predefined-schedule method) are novel. The release of the raw experimental data and usage of DACBench also fosters reproducibility of the claims, without having to repeat runs.

My assessment is as follows:

Strengths:
- Exhaustive study spanning different methods, models, datasets, and random seeds. It is refreshing to see a fair comparison of different methods, as usually each of the methods claims to be state-of-the-art.
- Clear contributions and an impact on the AutoML field are expected, especially for the results on LLMs

Drawbacks:
- While the paper focuses on the learning rate, the effect of the batch size is neglected. Related work has shown that learning rate and batch size influence each other (Smith et al.), thus, the results may be different at different batch sizes. While sweeping over different batch sizes to disentangle this effect could be computationally infeasible, I would have still expected a discussion of the problem and a small ablation study on some specific dataset and model combination.

Open Questions:
- In figure 4 b on page 7 the validation loss becomes worse for all methods up as the training progress. At the same time the validation accuracy further improves over training time. It is not clear to me why this is the case.

Minor:
- Missing space in line 152

In summary, I recommend a weak acceptance of the paper. While the study is relevant and exhaustive, the drawbacks and open questions should be addressed in a revised version of the paper.

(Smith et al.) https://openreview.net/pdf?id=B1Yy1BxCZ

**Review Confidence:**

4

**Review Rating:**

7

---

### Official Review · Reviewer_Sf9k · 2025-05-01

**Comments To Authors:**

## General Comments
This paper focuses on learning rate control in deep learning, a crucial aspect given the sensitivity of deep learning models to this hyperparameter. The authors compare multi - fidelity hyperparameter optimization (HPO), fixed - hyperparameter schedules, and hyperparameter - free learning approaches across various deep - learning tasks. They use SMAC3 with Hyperband for multi - fidelity HPO, four hyperparameter - free methods (COCOB, D - Adaptation, Prodigy, DoWG), and Cosine Annealing Warm Restarts (CAWR) as the scheduling approach. Experiments cover logistic regression (LIBSVM datasets), computer vision (CIFAR - 10, CIFAR - 100, DTD), and natural language processing (pre - training RoBERTa on BookWiki). The results show that no single method performs well across all tasks. They conclude that algorithm selection is important in learning rate control and suggest exploring methods like exposing key hyperparameters in hyperparameter - free learning and meta - learning to improve generalization.

## Pros
1. The paper conducts a thorough comparison of different learning rate control approaches, which is valuable for the community as it provides insights into the performance of these methods across diverse tasks.

2.  The suggestions for future research, such as exposing key hyperparameters in hyperparameter-free methods and using meta - learning, are practical and can potentially lead to more effective learning rate control strategies.

3. The authors provide detailed experimental settings, including dataset details, architectures, and hyperparameters, as well as code availability. This allows for easy replication of the experiments.

## Cons
1. By showing that no single method generalizes well across tasks, the authors highlight the need for algorithm selection methods in learning rate control. I do not fully agree with this view because the authors did not consider learning rate control based on dynamic algorithm configuration, which can achieve implicit automatic selection. I hope the authors provide more discussion on this point.

2. The experiments are limited to a specific set of datasets and models. It is unclear if the findings can be extended to other tasks, architectures, or datasets. The generalization of learned learning rate control policy is still unclear.

3. I hope to see more discussion on learning rate control in LLMs. For example, some articles consider linear decay first, followed by cosine decay [1]. I believe the community should pay more attention to these frontier issues in hyperparameter optimization for large model training, such as the scaling law in hyperparameter optimization [1,2].

[1] Predictable Scale: Part I — Optimal Hyperparameter Scaling Law in Large Language Model Pretraining. arxiv, 2025.

[2] A Multi-Power Law for Loss Curve Prediction Across Learning Rate Schedules. ICLR, 2025.

## Questions
1. Adam-family optimizers (such as AdamW) are currently the preferred optimizers for training LLMs. Besides the learning rate, beta also has a significant impact, and I hope to see some discussion on this.

**Review Confidence:**

4

**Review Rating:**

7

---

### Official Review · Reviewer_JVDF · 2025-05-01

**Comments To Authors:**

A nice paper showing the complexity of a priori selecting learning rate control strategies.

Questions:
- How do you say which method works best? Is it final result? If so, I think the claim in lines 205-206 is wrong, see e.g. iris, sensorless and vowel data sets.
- How does the validation loss and accuracy go up at the same time in figure 4?

Cons:
- Some of the colours in the plots are very hard to differentiate, e.g. Adam and COCOB in Figure 2.
- I think you use Adam and default interchangably. Can you be consistent so I'm sure?

Minor:
- Figure 2 comes before figure 1.

**Review Confidence:**

3

**Review Rating:**

8

---

### Official Review · Reviewer_yyks · 2025-05-06

**Comments To Authors:**

**Summary**

This paper presents a comparative study of different learning rate control paradigms in deep learning, including multi-fidelity hyperparameter optimization (HPO), fixed-hyperparameter schedules, and hyperparameter-free learning methods. The authors evaluate these approaches across a variety of deep learning tasks, spanning logistic regression, computer vision, and natural language processing. The core argument is that no single learning rate control method consistently outperforms others across all tasks.

**Strengths**

- The paper offers an extensive comparative analysis of different learning rate update methods, including fixed optimal schedules found via SMAC, learning rate schedule heuristics (e.g. cosine annealing) and hyperparameter-free learning rate update methods.
- The evaluation spans a good range of tasks with varying complexity, from logistic regression on LIBSVM datasets to computer vision tasks (CIFAR-10, CIFAR-100, DTD with ResNet and ViT architectures) and language model pre-training (RoBERTa on BookWiki).
- The motivation is clear and the authors highlight the "chaotic" nature of learning rate control methods and the lack of a one-size-fits-all solution. This justifies the call for more research into algorithm selection and meta-learning for this problem and proposes actionable and insightful future research paths.
- The authors provide their codebase and experimental setup, which is crucial for fostering reproducibility.

**Questions and Potential Improvements**

- While the paper reports which methods perform well on which tasks, a more in-depth analysis of why certain methods excel or fail in specific contexts could be beneficial.
- It would be interesting to also add in the set of LR controllers methods which exploit context information via LLMs, such as OptFormer (https://arxiv.org/pdf/2205.13320).
- Have the authors considered using DL optimizer benchmark libraries such as DeepOBS (https://github.com/fsschneider/DeepOBS)?
- The term "hyperparameter-free" is often a simplification, as these methods usually contain internal constants or design choices that can significantly influence performance, don't they?

Overall, the paper investigates a highly relevant and challenging problem in contemporary deep learning, therefore I recommend acceptance.

**Review Confidence:**

4

**Review Rating:**

8

---

### Meta-Review · Area_Chair_o7ou · 2025-05-08

**Recommendation:** Accept
**Confidence:** 4

**Metareview:**

The paper revisits various strategies for controlling the learning process in deep neural networks and presents a critical assessment of several methods from the literature.
All reviewers appreciated the thorough empirical evaluation across a diverse set of tasks.
Overall, I believe the paper could lead to interesting future research in the field of AutoML and recommend acceptance.